# Factors Influencing the Mental Health Consequences of Climate Change in Canada

**DOI:** 10.3390/ijerph16091583

**Published:** 2019-05-06

**Authors:** Katie Hayes, Peter Berry, Kristie L. Ebi

**Affiliations:** 1Dalla Lana School of Public Health, University of Toronto, Toronto, ON M5T 3M7, Canada; 2Climate Change and Innovation Bureau, Health Canada, Ottawa, ON K1P 5N7, Canada; peter.berry@canada.ca; 3Faculty of Environment, University of Waterloo, Waterloo, ON N2L 3G1, Canada; 4Center for Health and the Global Environment, University of Washington, Seattle, WA 98195, USA; krisebi@uw.edu

**Keywords:** climate change, mental health, adaptation, extreme weather, adaptive capacity, marginalized populations

## Abstract

Climate change is increasing risks to the mental health of Canadians. Impacts from a changing climate may outstrip the ability of Canadians and their health-sustaining institutions to adapt effectively and could increase poor mental health outcomes, particularly amongst those most marginalized in society. A scoping review of literature published during 2000–2017 explored risks, impacts, and vulnerabilities related to climate change and mental health. In this commentary, the authors present a new assessment of evidence from this scoping review and highlight factors that influence the capacity to adapt to the mental health consequences of a changing climate. Findings from this assessment reveal eleven key factors that influence the capacity to adapt: social capital; sense of community; government assistance; access to resources; community preparedness; intersectoral/transdisciplinary collaboration; vulnerability and adaptation assessments; communication and outreach; mental health literacy; and culturally relevant resources. Attention to these factors by Canadian decision makers can support proactive and effective management of the mental health consequences of climate change.

## 1. Introduction

Canadians are increasingly vulnerable to climate change impacts, including impacts to mental health. The term mental health is encapsulated in the definition of psychosocial health. Psychosocial health refers to psychological and social wellbeing [1]. The mental health outcomes related to climate change affect the way people think, act, feel, and interact, and they can arise from short-term hazards, such as flooding or hurricanes, or from an understanding of the potential long-term effects of climate change [1,2,3,4,5,6,7,8,9,10,11,12,13,14,15]. Mental health effects related to climate change include mental illness. Mental illness refers to conditions that result in moderate to severe mental disorders [15], and mental challenges [15,16] as well as positive mental health outcomes [15,17,18].

Recently, a scoping review by Hayes and Poland (2018) identified and synthesized evidence from the global literature on risks, impacts, and vulnerabilities related to climate change and mental health with a specific focus on the Canadian context [2]. Briefly, a scoping review refers to a type of literature review where researcher map evidence in a particular area of study, synthesize this evidence, and provide a narrative of the evidence, while also identifying gaps in the literature [19]. Notably missing in the global literature on climate change and mental health is a focus on psychosocial adaptation to a changing climate. This topic area is particularly salient in Canada, given it is warming twice as fast as the global average [20], thus both mitigation and adaptation efforts are urgently needed to protect human and planetary health. In this commentary, the authors present a new assessment of this original scoping review, exploring factors that influence the capacity to adapt to the mental health consequences of climate change in Canada. While the lessons learned from this commentary can be applied in other countries, this commentary focuses on the Canadian context to provide information required by decision makers facing challenges addressing climate change on mental health now. The objectives of this commentary are two-fold: (1) to identify response interventions and factors at the societal level that influence the capacity to adapt to the mental health consequences of a changing climate; and (2) to provide lessons learned for climate change adaptation in Canada. Briefly, societal level response interventions herein refer to: response interventions that can be taken by federal, provincial, municipal, or local decision-makers, to support adaptation within Canadian society.

Before delving into the topic area, it is prudent to define three key terms used throughout this document. First, response interventions refer to any form of formal or community-based care, policies, or programs that support mental health. Second, adaptation in this report refers to any mechanism, practice, or behavior-change that helps people, communities, and institutions cope with the mental health impacts of climate change and thrive, increasing their resilience to future impacts [18,21]. Third, adaptive capacity determinants refer to factors at the individual level or societal levels that support or enhance adaptation. At the individual level, adaptive capacity is determined by individual agency, the perceived need to adapt, the willingness to adapt, and the availability of resources to support adaptation [18]. At the societal level, adaptive capacity determinants include, for example: governance, economics, infrastructure, technology, information and skills, institutions, and equity [21]. The focus of this article is on societal level adaptive capacity determinants that support mental health in a changing climate.

## 2. Methods

In this commentary, we provide a new assessment of literature from a recent scoping review that explored the risks, impacts, and vulnerabilities related to climate change [2]. We review the evidence of climate change impacts to mental health and assess information that supports psychosocial adaptation to a changing climate in the Canadian context.

The original scoping review captured literature from peer-reviewed and grey sources. As research on climate change and mental health is relatively nascent, both peer-reviewed and grey literature were needed to understand the breadth and depth of climate change risks and impacts on mental health. Inclusion criteria for the original scoping review study included all English peer reviewed and grey literature published globally between 2000 and 2017 relating to climate change and mental health. Publications were included if they addressed: how a changing climate affects mental health (e.g., risks, impacts, and vulnerabilities) and/or how climate-related impacts to mental health are managed (e.g., responses and response capacity). Articles were excluded if they only included a passing mention that climate change impacts mental health or a passing mention about responding to the mental health consequences of climate change.

The original scoping review drew upon the global literature, applying lessons learned to a Canadian context. All types of literature were included in the scoping review search strategy, including: literature reviews, empirical studies, and reports. The original search terms included: “mental health” or “psychosocial” and “climate change”, and “risk” or “impact”, or “adaptation” or “response” or “resilience”, as well as synonyms and related words. The terms “resilience” and “global warming” were not explicitly included in the formal search terms; however, in the snowball search (highlighted below), the first author searched for literature that used these terms as well as “extreme weather”. The first author established the search strategy and conducted the literature review in consultation with the second author of this commentary. The following databases were used to search for peer reviewed literature: PubMed©, Scopus©, PsycINFO (Proquest)©, Cochrane Review©, and Google Scholar^TM^. The initial search captured 9079 articles. A snowball search identified a further 25 articles. Duplicates were removed and articles not meeting the inclusion criteria were removed after a review of all abstracts. A total of 276 articles met the inclusion criteria.

The original scoping review protocol was not registered; however, a summary of the literature reviewed for this scoping review—including the literature type, study design and location, participants, and outcome measures—can be found in the supplementary materials of the publication by Hayes and Poland (2018) [2]. The analyses performed by Hayes and Poland (2018) focused on the use of mental health indicators in climate change and health vulnerability and adaptation assessments in Canada. The new assessment presented in this commentary goes beyond the original study by highlighting factors that influence the capacity to adapt to the mental health consequences of climate change in order to support effective management of these consequences by Canadian decision-makers.

Analysis of the literature in this new assessment was informed by descriptive qualitative analysis of emergent themes pertaining to climate change and mental health adaptation. The first author of this paper conducted this analysis; the second and third authors provided their expert judgement on this analysis based on their experience in the field of climate change and health adaptation (see, e.g., [22,23,24,25,26,27,28,29]). As the objective of this commentary was to identify societal level factors that influence adaptation to the mental health consequences of climate change in Canada, the lead author conducted a descriptive qualitative analysis of emergent themes using a generic qualitative research method [30]. Briefly, a general descriptive method does not conform to any specific, established method (e.g., grounded theory) and is theoretically interpretively focused [31]. This interpretive analysis included a review of the literature for themes that pertain to the aforementioned societal level adaptive capacity determinants, including: governance, economics, infrastructure, technology, information and skills, institutions, and equity [21]. The first author reviewed the literature and made notes of reoccurring emergent themes pertaining to psychosocial adaptation and adaptive capacity in an Excel spreadsheet, the second and third author reviewed these emergent themes based on their expert judgement. These emergent themes are explored in Section 3.4.

## 3. Results

The majority of the articles in the original scoping review came from literature reviews (58%), followed by empirical research (42%), and finally by commentaries and grey literature (1%) [2]. Before delving into a discussion of the analysis from the reassessment of the literature, it is prudent to briefly highlight the key findings from the original scoping review as these findings enhance our understanding of climate change and mental health adaptation. There were three key findings: (1) there are a range of mental health risks and impacts from a changing climate; (2) there are unequal risks and impacts to people based on social and environmental factors; and (3) there are a range of response interventions that support mental wellness in a changing climate. These three core findings from the original scoping review are briefly presented followed by an in-depth discussion of factors that influence adaptation to the mental health consequences of climate change.

### 3.1. Mental Health Risks and Impacts of Climate Change

The literature reveals that the mental health outcomes related to climate change can include: post-traumatic stress disorder (PTSD), anxiety, depression, complicated grief and survivor guilt, recovery fatigue, and suicidal ideation from climate-related extreme weather events such as heatwaves, floods, hurricanes, and wildfires [3,4,5,6,7,8,9,10,11,12,14]. Other psychological impacts may include weakened social ties, increased stress levels, substance misuse, aggression, and violence related to resource scarcity caused by hazards such as rising heat levels, rising sea levels, and episodic droughts [11,12,13,14]. Emotions that affect mental health and can result in mental challenges, such as worry, anxiety, and feelings of impending doom related to the overarching awareness of climate change and the risks it poses to planetary and public wellbeing, are also of concern [6,7,8,9,10]. The mental health consequences of climate change may emerge immediately following an exposure to a climate change-related hazard or months to years later [9,32,33,34]. Further, while there are a host of psychological sequelae from climate-related extreme weather events and gradual climate change, people and communities with access to physical and psychological support can experience post-traumatic growth after an adverse event, which can give rise to feelings of optimism and altruism and foster compassion, generosity, a sense of meaning in a person’s life, and support broader community resilience [6,35,36]. However, the literature documenting positive mental health outcomes related to climate change is scant.

### 3.2. Unequal Risks and Impacts

The literature highlights that underlying the mental health consequences of climate change are issues of environmental justice that occur at the confluence of the Social Determinants of Health (SDoH) and the Ecological Determinants of Health (EDoH). The SDoH are factors that determine or influence health outcomes and health status, including employment, education, income, housing and working conditions, physical environments, social supports, access to healthcare, culture, gender, and childhood experiences [37,38]. The EDoH are environmental factors—such as climate change and atmospheric changes; ecotoxicity and pollution; and resource, ecosystem, and species depletion—that influence the capacity of health systems and individuals, and thereby affect health outcomes [39]. The SDoH and the EDoH help us understand that the mental health risks and impacts of climate change are unevenly distributed because of social and environmental factors, such as environmental racism, poverty, and social injustices.

### 3.3. Response Interventions

To better understand psychosocial adaptation opportunities to climate change events in Canada, it is valuable to recognize the current range of response interventions [3,6,11,14,33]. These include:policy responses that tend to enhance funding or access to mental health care;specific practices and behavioral interventions, such as inpatient or outpatient mental health care or counselling, and alternative mental health therapies such as mindfulness;community-based interventions, such as self-help groups, faith-based care or civic action groups;mental health care training, such as Psychological First Aid (PFA);awareness-raising of the mental health implications of climate change (e.g., via research dissemination and public health campaigns); andintegration of mental health care into disaster risk management plans.

All of these interventions may be specific to enhancing individual well-being, for example counselling services, or implemented at the societal level, such as policies that enhance access to services [3]. Interventions may be specific to addressing mental illness and challenges or enhancing positive mental health outcomes, such as environmental civic action groups that aim to reduce climate change impacts while fostering a sense of community, hope, meaning, and purposeful action [2]. Importantly, these interventions require actions by a variety of actors from various disciplines and these interventions operate across short- to longer-term time scales [3].

### 3.4. Exploring Psychosocial Adaptation Opportunities

As noted above, analysis in this reassessment of the scoping review literature included a qualitative analysis of emergent themes pertaining to climate change and mental health adaptation and expert judgement of climate change and health adaptation indicators. This analysis revealed that the effectiveness of currently available response interventions noted above is limited by an incomplete understanding of factors that can influence the capacity for adaptation by individuals and health decision makers. The key factors, herein referred to as “influencing factors”, that emerged in this reassessment of the literature include: social capital [40,41,42,43,44], sense of community [45,46,47,48,49], government assistance [11,13,14,36,40], access to resources [11,42,50,51,52,53,54], community preparedness [46,51,55], intersectoral/transdisciplinary collaboration [54,56,57,58], vulnerability and adaptation assessments [1,2,59,60], communication and outreach [8,46,61,62], mental health literacy [46,51,63,64], and culturally relevant resources [6,12,17,46,64,65,66,67]. These influencing factors can either support positive health outcomes when they are in place or when absent, act as barriers to psychosocial adaptation achievement and realization. Figure 1 is an original figure developed by authors of this paper to represent the connection between findings from the original analysis and the reassessment.

## 4. Discussion

The influencing factors identified in this research are discussed below along with examples (where applicable Canadian literature exists) drawn from the mental health response to Canada’s most recent, and costliest, disasters: the 2016 Fort McMurray, Alberta wildfires and the 2013 Southern Alberta flood [55]. Importantly, these influencing factors appear in no order of prioritization because knowledge of the importance of these factors does not allow for their prioritization and many work in combination to enhance psychosocial adaptation and resilience. Based on our analysis and awareness of climate change and health adaptations, it is our perspective that all of these factors in combination would optimize psychosocial adaptation in a changing climate.

### 4.1. Social Capital

In the literature on climate change and mental health, one of the most frequently highlighted factors that protects mental health and well-being is social capital. Social capital is inextricably linked with mental health, perhaps because social capital helps to reduce social isolation, loneliness, and feelings of abandonment [41,42,43,44]. Social capital can be described as: “the potential embedded in social relationships that enables residents to coordinate community action to achieve shared goals, such as adaptation to climate change” [41] (p. 502). Scholars note that social capital may outweigh economic assistance or assistance from aid groups or governments in post-disaster recovery and it underpins resilience [42,43]. LaLone [43] argued that community disaster planning and response organizations need to consider the role for social capital in supporting and enhancing recovery efforts. Dynes (as cited in [43]) suggested that the typical “command and control” style of government-led emergency response tends to overlook the significance and effectiveness of social capital in supporting recovery, and as such supporting and building social capital should figure as a central response strategy for governments responding to communities in distress.

### 4.2. Sense of Community

A sense of community is nested within the conceptualization and practice of social capital, and, similar to social capital, a sense of community (e.g., the feeling of togetherness and belonging) supports resilience in the face of a changing climate [45,46,47,48,49]. A sense of community and social support can play vital roles in psychological and psychosocial well-being, especially after a disaster. The catastrophic nature of disasters often results in displacement, fractured communication, and disconnection from homes and communities [46]. A key priority after a disaster is to rebuild community networks and to form new social connections [47,48]. A sense of community often requires strong leadership—either formal governmental leadership, or grassroots-led leadership, where leaders establish community trust, inclusivity, and foster empowerment [49].

### 4.3. Government Assistance

Government programs play a key role in providing and enhancing access to mental health care [11,13,14,36,40]. Mental illness in Canada is very costly; a recent report by the Canadian Mental Health Association estimated that 500,000 Canadians miss work every week due to mental health issues, costing the Canadian economy approximately $51 billion dollars per year [68].

Government programs can contribute to psychosocial well-being in the face of a changing climate through policy and program implementation or funding. In recent years, government assistance was integral in helping to address the mental health outcomes of climate-related events. The Province of Alberta, for example, responded to the 2013 Southern Alberta floods by allocating $50 million dollars to specifically support the current and future mental health needs of Albertans affected by the flood, along with the creation of the first Chief Mental Health Officer [54]. These funds were primarily used to deploy mental health practitioners from other cities and provinces to affected areas in Southern Alberta and to hire additional mental health specialists [54].

Government assistance is an important part of mobilizing mental health care to support psychosocial response in the aftermath of climate-related emergencies. As the timing of mental health outcomes vary and can linger for many years, and because of the increasing frequency of climate-related extreme weather events, mental health needs, including programs and interventions to protect people, are likely to grow.

### 4.4. Access to Care: Financial 

Psychosocial wellbeing in a changing climate may be thwarted by limited financial or physical access to mental health care [11,42,50,51,52,53,54,64]. Lack of access may stem from economics, geography, poor government planning, lack of trust, and/or poorly trained health professionals who may not recognize mental health symptoms or inappropriately triage patients with mental health needs.

While Canada has a universal, single-payer health care system, access to mental health care can still be impeded by financial constraints and availability of services depending on where an individual lives [50,68,69]. Mental health services vary by province and many mental health services are not covered under Canada’s universal health care program [50]. Services covered by universal health care in Canada include in-hospital mental health care and out-patient services from primary care physicians, nurses, and psychiatrists [50]. Patients seeking support outside of these professional domains will require private insurance to cover these expenses or they will need to pay for these services, which can make them inaccessible for many Canadians [50,69].

### 4.5. Access to Care: Physical 

Another key issue related to accessing mental health care is that during and following extreme weather disasters, mental health care facilities may be inaccessible due to infrastructure damage to buildings or roads to these facilities [11,53]. Further, mental health workers may not be able to reach health care facilities to provide care because of damage to their own property or to the roads they travel to reach care facilities, or due to personal injury [53]. In addition, some remote communities may not have (or may not have enough) mental health care facilities, resources, or practitioners prior to an extreme event. In such cases, mental health care is often introduced in a community post-event for a short period of time, however, long-term mental health care needs are often unmet [50].

### 4.6. Community Preparedness

In Canada, emergency management planning related to climate change has become an increasing area of concern. For example, Public Safety Canada recently released an environmental management strategy that specifically addresses the need for collaborative emergency planning and response (at all levels of government) related to climate change hazards [70]. Importantly, many preparedness plans and policies tend to overlook mental health care needs related to climate hazards. However, there are a few notable exceptions, such as the mental health and wellness recovery services guide to support Albertans affected by the 2016 wildfire; the guide targets residents, Alberta Health Services staff, physicians, and emergency responders [57]. This recovery plan was developed in an intersectoral fashion via consultation with: Indigenous community members, the Federal Department of Indigenous Services, Alberta Health Services Addictions and Mental Health, the municipality of Wood Buffalo, clients, and community members [57]. This plan targets the needs of Indigenous populations, outlines services that are specific to marginalized community members, highlights services available immediately after a wildfire, and describes the medium to long-term services in place [57]. Importantly too, the Canadian Psychological Association has crafted a list of mental health resources to support communities during and following emergencies, infectious disease outbreaks, and climate-related disasters (see [71]). Emergency preparedness planning that incorporates strategies to support mental health and wellbeing can provide communities with the tools needed to enhance psychosocial recovery following extreme weather-related events [46,51,55].

### 4.7. Intersectoral/Transdisciplinary Collaboration

A global review of the impacts of flooding on mental health during 2004–2010 states that: “a multi-sector approach that involves communities as well as agencies is the best way to promote wellbeing and recovery” [56] (p. 1). Intersectoral, as well as transdisciplinary or multi-sector collaboration occurs when people or groups from different disciplines work together to tackle complex issues. In the case of climate change and mental health in Canada, this may include collaboration amongst: front-line mental health workers (e.g., psychiatrists, psychologists, psychoanalysts, counsellors, social workers, primary care physicians, and community mental health care workers), faith-based and spiritual workers, emergency preparedness professionals, governments (at all levels), public health authorities, environmental and health NGOs, and climate and meteorological services [54,56,57,58]. 

### 4.8. Vulnerability and Adaptation Assessments

Climate change and health vulnerability and adaptation assessments provide information on how climate change affects health, describe populations most at risk, and outline specific actions to protect human health from climate change [1]. These assessments can support psychosocial wellbeing by detailing potential health risks and response opportunities at various levels (e.g., local level, provincial/state-level, and national level assessments) and amongst various institutions (e.g., corporations) [1].

In Canada, several climate change and health assessments have been completed at the municipal, regional and national levels, and amongst health units and acute care facilities (see [1,21,59,60]). However, mental health is often overlooked in these assessments because of the challenges related to climate change and mental health attribution and because of a lack of guidance on how to measure and monitor mental health effects related to climate change [2]. Notably, the next federal climate change and health assessment in Canada, to be released in 2021, will include a chapter detailing the climate change impacts to mental health [60].

### 4.9. Communication and Outreach

The American Psychological Association (APA) *Mental Health and Our Changing Climate* guidance document highlights the imperative role for frequent and clear communication to support psychosocial wellbeing in a changing climate [8]. The United Nations Office for Disaster Risk Reduction (UNISDR) suggests that clear, early-warning communication efforts can save lives during extreme weather events [61]. Furthermore, communication can increase community resilience by supporting people to assess risks and respond before an extreme weather event occurs [46].

Importantly, mass communication often relies upon technology so there is a need to build redundancy into technical systems (e.g., satellite communications, back-up generators, and door-to-door emergency service response communications) before, during, and after extreme events [46]. Other features of communication systems that are required to support and protect the psychosocial consequences of climate change-related extreme weather are communication systems from local and trusted sources, such as meteorologists [62].

### 4.10. Mental Health Literacy

Mental health literacy is defined as “the knowledge and beliefs about mental disorders which aid their recognition, management or prevention” [63]. Two of the main objectives of enhancing mental health literacy are to support help-seeking behaviors of people afflicted by mental health problems, and to support the education of people that mental health, similar to physical health, exists on a spectrum where people may experience affirmative mental health, or mental health problems, or mental illness. An additional component of mental health literacy is to reduce the stigma of mental health by bringing awareness of how common mental health issues are and that mental health problems and mental illness can afflict anybody at any time throughout a person’s life [63].

Mental health literacy is a skill that is needed amongst all health care professionals. An issue that emerged in New Orleans after Hurricane Katrina was the lack of training and skills of first aid responders and physicians in dealing with psychosocial and psychological trauma [46]; most first aid responders had to be trained in Psychological First Aid (PFA) in the initial weeks following the event [46,51]. Further, the stigma of mental health challenges and illness can act as a significant barrier to accessing mental health care. The majority of Hurricane Katrina respondents who did not seek mental health resources indicated that they wanted to handle the distress themselves; furthermore, they felt that there were negative social consequences to being labeled mentally ill or having mental health challenges [51]. One way of reducing the stigmatization of mental health challenges and illness is to increase mental health literacy amongst health care providers and the broader public. Psychological First Aid and Mental Health First Aid are two programs that can enhance mental health literacy.

### 4.11. Cultural Relevancy 

Cultural relevancy refers to response interventions that are culturally relevant and appropriate to all backgrounds and identities [48]. Culturally relevant response interventions that support mental health and wellbeing require that mental health practitioners and para-practitioners (e.g., first responders) exhibit cultural empathy [46]. Cultural empathy refers to the awareness and understanding of different cultural backgrounds, identities, and experiences of a person or group [46,48].

In Canada, there are concerns about the lack of cultural empathy when dealing with Indigenous health writ large, and more specifically in acknowledging the role for Indigenous Knowledge on the topic of climate change and health [66]. Indigenous peoples are “often viewed as powerless victims of climate change, overlooking how social, cultural, and economic conditions determine how climate change is experienced, understood, and responded to, and downplaying the accumulated knowledge and wisdom embodied in traditional knowledge systems that can provide valuable lessons for adaptation” [66]. These lessons are founded upon traditional understanding of lands, cultural identity, social networks and a holistic understanding of health [6,12,66]. Failing to recognize the valuable role that Indigenous Knowledge plays is a key barrier to adaptation and resilience to climate change, especially considering that Indigenous peoples are amongst the most at risk, particularly to the mental health impacts [6,12,67]. Further, cultural empathy goes beyond recognizing the valuable role of Indigenous Knowledge but also means a depth of understanding about the heterogeneity amongst Indigenous groups and thus a heterogeneity of how psychosocial adaptation is most effectively employed and by whom [17,18,66].

Indigenous Services Canada has specific programs on climate change and health in southern and northern Indigenous communities to increase resilience to impacts. [72]. There are numerous initiatives underway to engage with Indigenous communities and to integrate Indigenous Knowledge within climate change and health adaptation strategies, including on issues related to mental health impacts [72].

## 5. Conclusions

This commentary provides a reassessment of the scoping review conducted by Hayes and Poland in 2018 and a reflection on Canada’s costliest disasters (the 2016 Fort McMurray, Alberta wildfires and the 2013 Southern Alberta). In this commentary, the authors explore a number of factors that influence psychosocial adaption to a changing climate in Canada: social capital, sense of community, government assistance, access to resources, community preparedness, intersectoral/transdisciplinary collaboration, vulnerability and adaptation assessments, communication and outreach, mental health literacy, and culturally relevant resources. Importantly, our analysis is limited to key influencing factors that we deemed most applicable to a Canadian context. Absent from the analysis is a focus on factors that would be, for example, most relevant to low- and middle-income countries. While the analysis is centered on a Canadian context, these findings may be useful to decision makers in other jurisdictions and countries facing similar challenges responding to climate change impacts on mental health.

Climate change adaptation policies may benefit from knowledge of the mental health risks associated with a changing climate, resources required to reduce these risks, and of key factors that can influence the magnitude and pattern of mental health outcomes. Health authorities can obtain this information through vulnerability and adaptation assessments, and in collaborations with partners to reduce the health consequences of climate-related hazards. Further, this information can be used by health care providers to support individuals and communities to more effectively manage mental health issues associated with climate variability and change.

## Figures and Tables

**Figure 1 ijerph-16-01583-f001:**
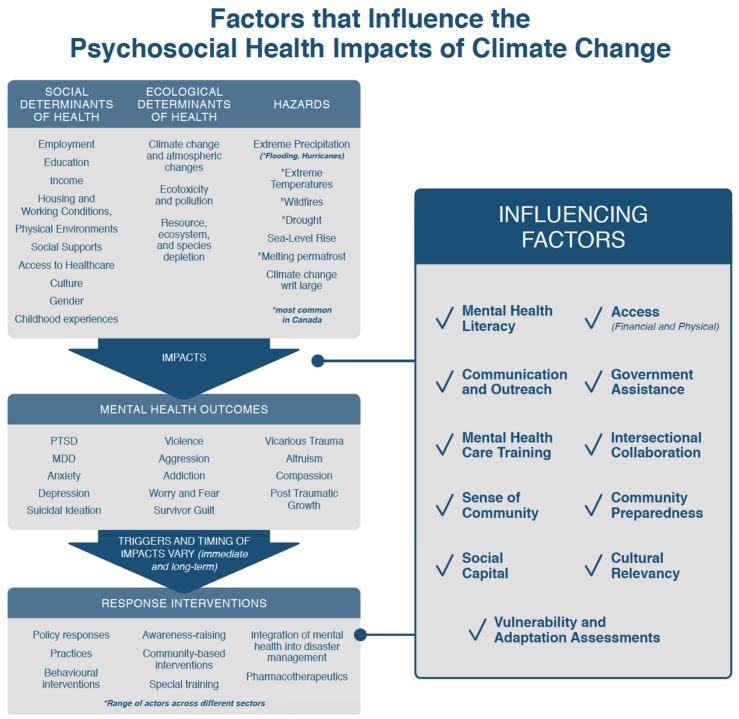
Factors that influence the psychosocial health impacts of climate change. A framework showing the mental health consequences of climate change and how these consequences are mediated by the social and ecological determinants of health, response interventions, and factors that influence psychosocial adaptation when they are in place or when absent act as barriers to psychosocial adaptation.

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
