# Peer review of "Factors Influencing the Mental Health Consequences of Climate Change in Canada"

_ijerph, 2019, doi:10.3390/ijerph16091583_

Round 1

Reviewer 1 Report

This article provides a reasonable summary of the current literature on the topic of mental health issues relevant to climate adaptation efforts. I would like to see a paragraph in the introduction that describes what is unique about Canada that makes this relevant only to the Canadian context.

It generally appears honest and insightful in its qualitative analysis of emergent themes although I’d like to see a more extensive description of how analyses were conducted. 

I would like to see the conclusion section expanded to articulate the limitations inherent to this form of analysis. Are there areas where the literature is absent altogether but that should be considered?  It is also important to discuss whether the conclusions are generalizable beyond the Canadian context. 

The writing is somewhat repetitive in places and could be streamlined in that respect. The article is otherwise well-written and appropriately brief.

Author Response

On behalf of Dr. Ebi, Dr. Berry, and myself thank you for your comments. Please find attached a word document that includes point-by-point responses (highlighted in yellow) to your comments on our submission titled, Factors Influencing the Mental Health Consequences of Climate Change in Canada.

Reviewer 2 Report

This manuscript describes additional findings related to a scoping review on climate change and mental health that was previously detailed within the article “Addressing Mental Health in a Changing Climate: Incorporating Mental Health Indicators into Climate Change and Health Vulnerability and Adaptation Assessments”, which was published in IJERPH on August 22, 2018. The current effort was developed to “present a new assessment of this original scoping review, exploring factors that influence the capacity to adapt to the mental health consequences of climate change in Canada”.

Identifying such factors would certainly make a contribution to the literature and could help inform some of the ongoing policy-related efforts described within this manuscript, particularly the Canadian federal climate change and health assessment report set to be released in 2021. A robust framework of influencing factors, such as that initiated in Figure 1, would be a particularly valuable addition to the literature, as would a thorough exploration of factors specific to the Canadian context, such as the impact of cultural relevancy included in Section 4.11.

Unfortunately, the current manuscript has a number of major methodological flaws that prevent it from making this contribution in its current state. Most importantly, it appears to have been submitted as an original research manuscript, but the underlying scoping review has already been published. In addition, the qualitative appraisal methods that form the heart of the effort are insufficiently described, precluding appraisal of their suitability and robustness for use in a research study.

I believe resolving these flaws so the manuscript would fulfill the requirement “that the work reports scientifically sound experiments and provides a substantial amount of new information” (as noted in the IJERPH instructions for authors) would require such a substantial change to the research design that the manuscript must be rejected in its current form. However, I believe that it could be restructured into a commentary and have therefore provided some specific suggestions for improvements below. Such a commentary could maintain the description of the influencing factors identified as part of the original study and their connections to specific Canadian events such as the 2013 Southern Alberta floods and the 2016 Fort McMurray wildfires, as well as the material on cultural relevancy, which is a strength of the current submission.

Introduction
This section could be strengthened by the inclusion of additional citations throughout, particularly for lines 28-33. In addition, it would benefit from the creation of a new paragraph describing the gap in the evidence base the manuscript is intended to fill, as well as the justification for focusing on Canada, rather than another setting.

Methods
Should the authors choose to resubmit the manuscript as a scoping review, it would be necessary to expand upon the initial search to include a new construct with terms specific to the new focus, such as “adaptive capacity”, “intervention”, and “management”. In light of the fact that less than 15% of the studies included in the supplementary material for the original report were conducted in Canada, I would also recommend the inclusion of an additional construct and revised inclusion criteria to restrict studies to this setting. Although registering a protocol isn’t a requirement for a scoping review, additional details on the search strategy, inclusion/exclusion criteria, and justification for the removal or addition of articles at each stage of the review process should still be provided as supplementary material, along with a summary table of the type included with the earlier scoping review. Finally, the exact nature of the “descriptive qualitative analysis of emergent themes” should be clarified, with reference to a specific methodological framework, tools, and any software utilized.

Results
Currently, more than a third of this section (lines 105-168) is meant to “briefly highlight the key findings from the original scoping review”. However, these are not original results, so devoting such a substantial proportion of this section to this material seems inappropriate. This content could be replaced with a new summary of the findings from an update scoping review or substantially condensed. In terms of the new material, Section 3.4 describes a number of factors that limit the effectiveness of current interventions, but exactly how the “analysis revealed” these factors is unclear, and many of them are supported by just a single study, often one not carried out within Canada. Figure 1 would be a key component of either a novel scoping review or a commentary, but it could be improved by some distinction between those factors that “support positive health outcomes” and those that “act as barriers”.

Discussion
This section begins with an introductory paragraph rightly highlighting the limitations of existing knowledge regarding the relative importance of and interrelationships among specific influencing factors. However, this statement should be complemented by some insights into the authors’ perspectives on these issues, even if brief. There are also a number of subsequent sections that could be improved by such perspectives, particularly 4.1 and 4.10. Finally, many statements contained within this section may be supported by the literature, but are insufficiently cited, particularly those made on lines 229-233, 236-238, 260-262, 280-285, and 295-297 (which currently only cites the prior scoping review).

Conclusions
This section does not seem to reflect the specific findings of the current manuscript, particularly the identified influencing factors and connections to specific events such as the Southern Alberta floods and Fort McMurray wildfires. 

Author Response

(The authors gave the same response as above.)

Reviewer 3 Report

Thank you for the opportunity to review this manuscript of the important topic of climate change and mental health adaptation. 

I few comments regarding this manuscript:

Line 157: are self-help groups included in community-based interventions?  Also would there be any place for alternative mental health therapies in this section regarding resources? ie: Mindfulness etc.

Page 5 Factors that influence Psychosocial impacts of Climate Change.

Any mention of solastagia – anxiety related to environmental change in the literature?

Page 5 Last line in Discussion – consider stating”…enhancing psychosocial adaptation and resilience.”

Lines 261-267: Does Canada have mental health supports for disasters? It seems that it would be important to include what is already available for disasters regardless of climate related disasters.

Line 306: remove “relative” in front of “risk” as it implies a statistical measure.  Just use “risk.” 

Line 341: “role that Indigenous Knowledge plays is a key barrier to adaptation and resilience to climate change, especially….

Conclusion: consider adding the need to address climate mental health mitigating factors and resilience in the conclusion.

Author Response

(The authors gave the same response as above.)

Round 2

Reviewer 2 Report

I appreciate the close attention the authors paid to the feedback provided on their initial submission, as captured in their point-by-point response memo. The shift from an original research manuscript to a commentary is perhaps the largest revision, and I believe this change resolves the major methodological flaw identified in my initial review. In addition, the vast majority of the other salient comments have been thoroughly addressed, greatly strengthening the overall manuscript.

However, I was unable to locate a number of the changes described in the response memo within the revised manuscript, in part because some of the page number/line location combinations cited appear not to exist (e.g., “Page 4, Lines 203-207”, “Page 6, Lines 293-295”, “section 4.1 Page 6, Lines 304-310”, “section 4.10 Page 8, Lines 452-457”) and in part because some sections where changes are noted in the memo appear to be identical in both versions of the manuscript (e.g., the authors state that an “update has been made to briefly note how these [influencing] factors support or act as barriers to health outcomes (see Page 4, Line 193)”, but neither the figure nor caption seem to have been revised). In addition, some of the newly integrated content could benefit from a thorough copy-edit, particularly to avoid the construction of unnecessary compound nouns (e.g., “Canadian-context”, “societal-level”) and the misuse of semi-colons in place of commas in simple lists.

I am confident that these relatively minor concerns can be addressed by the authors as part of a final read-through of the revised manuscript, and am therefore recommending that the manuscript be accepted after minor revision.

Author Response

Thank you kindly for your review and comments. Please find attached our responses to your feedback.
